# Enhanced Hydrophilic and Electrophilic Properties of Polyvinyl Chloride (PVC) Biofilm Carrier

**DOI:** 10.3390/polym12061240

**Published:** 2020-05-29

**Authors:** Haifeng Cai, Yang Wang, Kai Wu, Weihong Guo

**Affiliations:** Polymer Processing Laboratory, Key Laboratory for Preparation and Application of Ultrafine Materials of Ministry of Education, School of Materials Science and Engineering, East China University of Science and Technology, Shanghai 200237, China; 18721358176@163.com (H.C.); m18864832205@163.com (Y.W.); 13122320038@163.com (K.W.)

**Keywords:** polyvinyl alcohol, cationic polyacrylamide, polyvinyl chloride, azodicarbonamide

## Abstract

Polyvinyl chloride (PVC) biofilm carrier is used as a carrier for bacterial adsorption in wastewater treatment. The hydrophilicity and electrophilicity of its surface play an important role in the adsorption of bacteria. The PVC biofilm carrier was prepared by extruder, and its surface properties were investigated. In order to improve the hydrophilicity and electrophilic properties of the PVC biofilm carrier, polyvinyl alcohol (PVA) and cationic polyacrylamide (cPAM) were incorporated into polyvinyl chloride (PVC) by blending. Besides, the surface area of the PVC biofilm carrier was increased by azodicarbonamide modified with 10% by weight of zinc oxide (mAC). The surface contact angle of PVC applied by PVA and cPAM at 5 wt %, 15 wt % was 81.6°, which was 18.0% lower than pure PVC. It shows the significant improvement of the hydrophilicity of PVC. The zeta potential of pure PVC was −9.59 mV, while the modified PVC was 14.6 mV, which proves that the surface charge of PVC changed from negative to positive. Positive charge is more conducive to the adsorption of bacteria. It is obvious from the scanning electron microscope (SEM) images that holes appeared on the surface of the PVC biofilm carrier after adding mAC, which indicates the increase of PVC surface area.

## 1. Introduction

With the continuous development of industry and the continuous improvement of people’s living standards, more and more garbage has been produced, which has led to an increase in sewage discharge. The generation of various pollutants in the aquatic environment has become an issue of increasing global concern in the past few decades [1]. Wastewater treatment plants are designed to eliminate various chemical and microbial pollutants in wastewater [2]. Due to its low cost and high treatment efficiency, biological wastewater treatment processes remain the most widely used method for removing organic pollutants and nutrients [3,4,5]. Generally, biofilm-based wastewater treatment systems have several advantages: for example, their high active biomass concentration, short hydraulic residence time, low space requirements, and less sludge production. In particular, the microbial communities in biofilms are diverse, which allow degrading a wide range of organic pollutants [6]. Meanwhile, the attachment and formation of biofilms largely depend on the surface properties of the biofilm carrier, including the physical/chemical properties of the carrier surface, the charge properties of the carrier surface, and the surface roughness of the carrier and so on [7,8,9,10,11,12]. In recent studies, the hydrophilicity and electronegativity of the carrier surface play an important role in the formation and treatment efficiency of biofilms [13,14,15]. Therefore, the selection of the required carrier is considered to be the decisive factor affecting bacterial adhesion and biofilm formation [16,17].

The interaction between bacteria and the surface of the carrier is mainly influenced by interface interactions such as repulsive force/attraction and Van der Waals force [18]. It is widely believed that the cell wall surface of bacteria contains functional groups such as –OH, –COOH, and –CHO. Hydrogen bonds can be formed between the carrier surface and bacteria [18,19]. In earlier research, it was found that the adhesion of bacteria to hydrophobic surfaces is significantly reduced [20,21]. In addition, the microbial surface is negatively charged as phosphoric acid and carboxylic acid groups in the microbial cell membrane. Therefore, the electrophilic property of biofilm carriers has a great effect on the adhesion of microorganisms and the formation of biofilms [22,23,24].

An ideal biofilm carrier should have the following characteristics: low cost, excellent mechanical strength, low density, stability, large specific surface area, high bioaffinity, anti-biodegradability, and anti-aging [25,26,27]. Polyvinyl chloride (PVC) owns most of the advantages described above. PVC is an important thermoplastic that can be used in a wide range of applications such as pipes, profiles, bars, films, insulation materials, etc. Moreover, PVC has excellent properties such as non-flammability, corrosion resistance, insulation, and wear resistance. The most prominent advantage of PVC is its low price, higher tensile strength, and larger bending strength [28]. However, the bioaffinity of PVC is relatively low, and its hydrophilicity is relatively weak. At the same time, its surface is relatively smooth, and its specific surface area is relatively small. Moreover, the surface of pure PVC is negatively charged, which is the same as the surface of bacteria [6,29,30,31]. These disadvantages make it difficult for bacteria and microorganisms to attach to PVC biofilm carriers.

In this study, in order to solve the above-mentioned shortcomings of PVC, such as weak hydrophilicity, small surface area and negatively charged surface, a typical hydrophilic polymer PVA was incorporated into the PVC to improve its hydrophilicity. In order to change the surface chargeability of the PVC biofilm carrier, positively charged polymer cationic polyacrylamide (cPAM) was incorporated into the PVC. At the same time, a compound blowing agent azodicarbonamide (mAC) was added to increase the specific surface area and surface roughness of the PVC biofilm carrier.

## 2. Experimental

### 2.1. Materials

PVC (SG-5) was purchased from Yuyao Maiduo Plastic Chemical Co., Ltd. (Yuyao, China). PVA and Stearic acid were purchased from Shanghai Lingfeng Chemical Reagent Co., Ltd. (Shanghai, China). cPAM was purchased from Zhengzhou Jintai Environmental Protection Technology Co., Ltd. (Zhengzhou, China). Azodicarbonamide was purchased from Shanghai Tengzhun Biological Technology Co., Ltd. (Shanghai, China). Calcium zinc stabilizer was purchased from Guangdong Winner New Material Technology Co., Ltd. (Foshan, China). Dioctyl terephthalate (DOTP) was purchased from Jining Baichuan Chemical Co., Ltd. (Jining, China). Zinc oxide was purchased from Sinopharm Chemical Reagent Co., Ltd. (Shanghai, China). Antioxidant 1010 was purchased from Shanghai Xian Ding Biological Technology Co., Ltd. (Shanghai, China). The specific information of the materials is shown in Table 1.

### 2.2. Preparation of PVC Biofilm Carrier

PVC resin (100 phr) was mixed with 0–10 wt % of PVA and 0–15 wt % of cPAM (the weight ratios of PVC to PVA were 100/0, 100/5, 100/10 and the weight ratios of PVC to cPAM were 100/0, 100/3, 100/6, 100/9, 100/12, and 100/15) using calcium zinc stabilizer (8 phr) as a heat stabilizer, DOTP (60 phr) as a plasticizer, and mAC (1 phr) as a chemical blowing agents. Then, stearic acid (0.4 phr) and antioxidant 1010 (0.5 phr) were added, and the mixture was mixed thoroughly in a high-speed mixer (SHR-10A, 750 r/min). After the mixture was well mixed and the DOTP was fully absorbed by the mixture, the mixture appears loose. Then, the mixture is taken out and dried. Then, the mixture was extruded and winded in a twin-screw extruder (18 °C, 15 r/min). As a control, PVC without fillers was prepared following the same procedure [28,32]. The preparation of PVC biofilm carrier and the photo of sample are shown in Figure 1 and Figure 2.

The formulation and abbreviation of the samples are listed in Table 2.

### 2.3. Surface Contact Angle

The surface contact angle of the resultant PVC biofilm carriers was measured by a Standard type contact angle meter (JC2000D2, Shanghai Zhongchen Digital Technology Equipment Co., Ltd., Shanghai, China). Distilled water was slowly dropped onto the surface of the specimens. Then, a photograph of the water droplets and the surface of the specimen were taken by the contact angle meter. At least five different locations were measured for each specimen. The conductivity of the used distilled water was 0.056 μS/cm, and the resistivity of the used distilled water was 18.2 MΩ·cm at 25 °C. The indoor temperature is 25 ± 0.5 °C, and the indoor humidity is 50% ± 1% during the test.

### 2.4. Fourier Transform Infrared Spectra

Fourier transform infrared (FTIR) spectra were measured on a NICOLET 6700 spectrometer (Thermo Scientific Co., Waltham, MA, USA) from 4000 to 400 cm^−1^ to study the interaction between PVA and PVC. The samples tested are in powder form, and potassium bromide pressed-disk technique was used.

### 2.5. Zeta Potential

The zeta potential of the specimens was measured by a laser particle size analyzer (LS230, Microtrac Inc., Clay, FL, USA). First, we dissolve the powder sample in distilled water and place it in the sample cell for measurement. The conductivity of the used distilled water was 0.056 μS/cm and the resistivity of the used distilled water was 18.2 MΩ·cm at 25 °C.

### 2.6. Field Emission Scanning Electron Microscope

The surface morphology of the specimens was observed by using an S-4800 field emission scanning electron microscope (Hitachi, SEM, Tokyo, Japan). Prior to SEM observation, all specimens were coated with a thin gold layer. The magnification power of SEM was 1.00k and 2.00k, and the voltage was 15.0 kv.

### 2.7. Mechanical Properties

The tensile properties were determined by using an MTSE44 universal testing machine (Jinan Yongce Industrial Equipment Co., Ltd., Jinan, China) in accordance with ISO 527 respectively. At least five independent measurements were conducted for each sample (75 mm × 4 mm × 2 mm for tensile test).

## 3. Results and Discussion

For PVC biofilm carriers prepared by different formulations, their surface contact angle, zeta potential, mechanical properties, and surface morphology were tested by using the corresponding equipment.

### 3.1. Hydrophilicity of PVC Biofilm Carrier

The hydrophilicity of the PVC biofilm carrier plays an important role in the adsorption of bacteria. Since the cell wall surface of bacteria contains functional groups such as –OH, –COOH, and –CHO, it prefers to be adsorbed on the surface of strong hydrophilic carriers [18,19]. The surface contact angle can be used intuitively to characterize the hydrophilicity of the carrier. In order to investigate the modification effect of polyvinyl alcohol on polyvinyl chloride, the surface hydrophilic properties of polyvinyl chloride were tested and characterized. The surface contact angle of unmodified PVC and modified PVC were measured by a standard-type contact angle meter. Figure 3 and Figure 4 are the surface contact angle of the PVC biofilm carrier. It shows the change in the surface contact angle of the specimens after the addition of PVA (PVC/PVA-5, PVC/PVA-10) and mAC (PVC/mAC-PVA, PVC/mAC-PVA-5, PVC/mAC-PVA-10).

Compared with pure PVC, the contact angle of the PVC sample with 5 wt % of PVA has been significantly reduced, from 99.5° to 84.2°, which is reduced by 15.4%. The significant reduction of surface contact angle indicates that PVC was successfully modified by PVA. The specific reaction mechanism is shown in Figure 5. PVA is an aqueous polymer whose segment is rich in hydroxyl groups. When it is added to PVC, the hydroxyl groups on the segments partially interact with chlorine on the PVC to form hydrogen bonds. This makes PVA well integrated on PVC. At the same time, the hydroxyl group that exists on PVA alone improves the hydrophilicity of PVC. Hydroxyl is a hydrophilic group. The improvement of the hydrophilicity of PVC is due to the independent hydroxyl group on PVA. The more hydroxyl groups on the modified PVC, the better its hydrophilic property and the lower the surface contact angle. When the amount of PVA was increased to 10 wt %, the surface contact angle of the sample increased from 84.6° to 86.4°. This could be because the amount of PVA added is too large, causing the hydroxyl group between PVA to act to generate hydrogen bonds, which in turn reduces the binding of PVA to PVC and the number of PVA independent hydroxyl groups. Therefore, it is more appropriate to control the amount of PVA added to 5 wt %.

Bacteria and microorganisms used to treat sewage are generally hydrophilic [18,20,21]. The hydrophilicity of PVC biofilm carrier modified with PVA was effectively improved. Therefore, bacteria and microorganisms can be more effectively attached to the modified PVC biofilm carrier. The increase in the amount of bacteria on the biofilm carrier can effectively improve the ability and efficiency of sewage treatment.

At the same time, as shown in Figure 4, after the addition of mAC, the change trend of the contact angle of the sample is consistent with the above description. This trend is expected. The role of the mAC is to increase the surface area of the PVC biofilm carrier by foaming and has no effect on the surface chemistry of the sample. Neither PVC nor PVA will react with mAC, so the chemical properties of the modified PVC will not be affected. This result shows that there is no side effect when adding PVA and mAC into PVC at the same time.

Figure 6and Figure 7 are the surface contact angle of the PVC biofilm carrier, which contains different amounts of cPAM. It shows the change in the surface contact angle of the sample after adding cPAM based on the addition of 5 wt % PVA and 1 wt % mAC. It can be seen that compared with pure PVC, after the amount of cPAM added reaches 12 wt %, the contact angle of PVC is further reduced to 80.8°. The data indicate that compared with the PVC biofilm carrier modified by PVA, the hydrophilic property of the PVC biofilm carrier added with a certain amount of cPAM and PVA is stronger. The reason for this phenomenon is that the amino group on the cPAM chain segment plays a role in enhancing the hydrophilicity of the PVC biofilm carrier. The mechanism between cPAM and PVC is similar to PVA and PVC. Although the polarity of the amino group is not as large as that of the hydroxyl group, after the amount reaches a certain amount, the amino group can also play a role in giving the PVC a certain hydrophilicity.

### 3.2. FTIR Spectra and Energy Dispersive Spectrum (EDS)—The Change of Hydroxyl Groups

In the above part, the change of the surface contact angle of the carrier indicates that the hydrophilicity of the carrier is enhanced after the addition of PVA. That is because the hydroxyl group that exists on PVA alone improves the hydrophilicity of PVC. The specific reaction mechanism is shown in Figure 5. In order to verify that the improvement of the hydrophilic property of the PVC biofilm carrier is indeed due to the reaction between PVA and PVC, the changes in surface functional groups and elements of the PVC before and after modification need to be tested and investigated. The FTIR and EDS spectra of modified PVC and unmodified PVC were tested and characterized. Figure 8 is the FTIR spectra of unmodified PVC (a) and PVC/PVA-5 (b). The purpose of testing the infrared spectrum of PVC is to investigate whether PVA is effective in modifying PVC. As shown in Figure 8, the 2959 cm^−1^ absorption peak of PVC and the 2958 cm^−1^ absorption peak of PVC/PVA-5 are the stretching vibrations of CH2. The 1462 cm^−1^ absorption peak of PVC and the 1459 cm^−1^ absorption peak of PVC/PVA-5 are the bending vibration of CH_2_. Meanwhile, the absorption peaks of 1265 and 1272 cm^−1^ are attributed to the wobbling vibration of the adjacent carbon atom of CH_2_, which is connected to a chlorine atom. The 1724 cm^−1^ absorption peak of PVC/PVA-5 is the absorption peak of C=O remaining in PVA. Besides, in Figure 8, the absorption peak around 3400 cm^−1^ indicates the association absorption peak of O–H [28,33]. Compared with unmodified PVC, the absorption peak of O–H of PVC modified by PVA is obviously sharper and wider. It shows that a certain amount of O–H appears in the surface of the modified PVC. This result proves that the reaction mechanism between PVA and PVC that we mentioned earlier is reasonable and correct.

For further verification, EDS analysis was performed on the samples. The EDS results of PVC and PVC/PVA-5 are summarized in Figure 9 and Table 3.

It can be seen from Table 3 that the element content of the unmodified PVC is C (45.4 wt %), O (8.4 wt %), and Cl (46.2 wt %), while the element content of the PVC modified by PVA is C (63.7 wt %), O (17.5 wt %), and Cl (18.8 wt %). Compared with unmodified PVC, the content of oxygen and carbon of the modified PVC increased significantly, while the content of chlorine decreased significantly. This indicates that PVC successfully interacts with PVA, resulting in an increase in the content of oxygen and carbon element on the surface.

Moreover, in order to evaluate the miscibility of PVC and PVA in the present samples, the melting characteristics of modified PVC and unmodified PVC were tested and characterized. Figure 10 is the melting characteristics of unmodified PVC, PVA and PVC/PVA-5. It can be seen from Figure 10 that the melting points of unmodified PVC, PVA, and PVC/PVA-5 are 158.1, 192.7, and 170.6 °C. The melting point of PVC/PVA-5 is between unmodified PVC and PVA. Moreover, it can be clearly seen from the melting curve of PVC/PVA-5 in Figure 10 that only a relatively smooth melting peak appears in PVC/PVA-5, and no other peaks appear. This shows that PVC and PVA are well miscible together, and their miscibility is relatively good.

From the melting characteristics of PVC, PVA, and PVC/PVA-5 in Figure 10, it can be analyzed that the miscibility of PVC and PVA is relatively good. Combining the results of FTIR spectra and EDS spectra, it can be determined that PVC is successfully modified by PVA. The hydrophilic property of PVC is effectively enhanced.

### 3.3. Electrophilicity of PVC Biofilm Carrier

The microbial surface is negatively charged as phosphoric acid and carboxylic acid groups in the microbial cell membrane. Therefore, the electrophilic property of biofilm carriers has a great effect on the adhesion of microorganisms and the formation of biofilms [22,23,24]. In order to change the surface chargeability of the PVC biofilm carrier from negative to positive, positively charged polymer cationic polyacrylamide (cPAM) was incorporated into the PVC. The zeta potential can be used intuitively to characterize the electrophilicity of the carrier. In order to investigate whether the surface charge property of PVC have changed after the addition of cPAM, the charge properties of the surface of the PVC biofilm carrier were tested and characterized. The zeta potential of modified PVC and unmodified PVC were measured by a laser particle size analyzer.

Figure 11 is the zeta potential of the PVC biofilm carrier. It shows the change in zeta potential of the sample after adding different amounts of cPAM (0, 3, 6, 9, 12 and 15 wt %).

It can be seen that as the amount of cPAM added increases, the zeta potential of the sample also increases from −9.59 mV for pure PVC to 14.6 mV for 15 wt %. This shows that after the addition of cPAM, the surface chargeability of PVC gradually changes from negative to positive, and as the amount of cPAM increases, the zeta potential increases, indicating that the positive charge has also become stronger and stronger. Figure 12 shows the electrostatic interactions between unmodified PVC/modified PVC and bacteria. As shown in Figure 12, bacteria are generally negatively charged in water, and unmodified PVC is also negatively charged. There is a repulsive effect between the two, which is not conducive to the adhesion of bacteria. Meanwhile, the modified PVC has a positive charge, which will attract the bacteria and facilitate the adsorption of bacteria. The increase in the amount of bacteria on the biofilm carrier can effectively improve the ability and efficiency of sewage treatment [15,30,34].

### 3.4. Surface Morphology of PVC Biofilm Carrier

In order to investigate the surface change of PVC after the addition of the compound chemical blowing agent mAC, the surface morphology of the PVC biofilm carrier before and after foaming was observed by S-4800 field emission scanning electron microscope. Figure 13 is the SEM images of the surface morphology of pure PVC (Figure 13a), 1 wt % mAC (Figure 13b), 5 wt % PVA (Figure 13c), and 1 wt % mAC with 5 wt % PVA (Figure 13d). The part circled by the red circle in Figure 13 is the hole. In a field emission scanning electron microscope, the magnification power of SEM was 2.00 k. Comparing Figure 13a,b, it can be seen that the surface of pure PVC is relatively smooth before the addition of mAC. After the addition of mAC, the surface of the PVC sample clearly shows holes, which indicates that mAC has successfully foamed. Thereby, the surface area of the sample was increased. Although the distribution of pores is relatively irregular, it can be seen from the figure that the size of the pores is more than the micron level. The general diameter of bacteria is about 1 μm, so these holes are completely satisfied for the adhesion and growth of bacteria on it. The principle of mAC is that at the processing temperature of PVC, zinc oxide will activate azodicarbonamide and reduce its decomposition temperature, so that its decomposition temperature is close to the processing temperature. Therefore, mAC will decompose during the processing of PVC. Its decomposition products are mainly harmless gases, such as nitrogen, which will escape and form pores on the surface of the material [35]. Comparing Figure 13c,d, after adding 5 wt % PVA and 1 wt % mAC, the formation of the pores on the surface of the sample is consistent with the one described above.

Figure 14 is the SEM surface morphology of the PVC biofilm carrier, which contains different amounts of cPAM based on the addition of 5 wt % PVA and 1 wt % mAC. It shows the SEM surface morphology of pure PVC (Figure 14a), 3 wt % cPAM (Figure 14b), 12 wt % cPAM (Figure 14c), and 15 wt % cPAM (Figure 14d). The part circled by the red circle in Figure 14 is the hole. At the same time, as shown in Figure 14, the individual holes in Figure 14a,c are enlarged, and the enlarged images of the holes are indicated by arrows. The SEM magnification of the enlarged image of the hole is 10.0 k. From these SEM images, we can find that the amount of cPAM added will not affect the foaming on the basis of adding the blowing agent mAC. The blowing agent can still foam normally to form pores on the surface of the sample, thereby increasing the surface area of the sample.

### 3.5. Mechanical Properties

Figure 15 and Figure 16 is the tensile strength of the PVC biofilm carrier. It shows the tensile properties of each sample after adding PVA, mAC, and cPAM. It can be clearly seen from the figure that compared with pure PVC, after adding PVA, mAC, and cPAM, the tensile properties of the sample did not change much, the highest was 5.6%, and the lowest was 4.2%. This is because the added substances did not change the structure and segment of the PVC matrix. The main change was the surface properties of PVC. The most important effect was the combination of hydrogen bonding with PVC, and no other chemical reaction. The tensile properties of PVC are mainly imparted by its structure. Therefore, the tensile properties of the samples do not change greatly on the basis that the main structure and the segment of the PVC matrix are not greatly changed. This result illustrates that the amount of each filler added is appropriate [36,37,38,39,40].

## 4. Conclusions

In this study, PVA and cPAM were incorporated into the PVC matrix. At the same time, mAC was also added. Then, the surface properties and zeta potential of the PVC biofilm carrier were investigated. The results showed that the surface contact angle of the PVC biofilm carrier was significantly lower than that of pure PVC after the addition of PVA and cPAM. This is the effect of the hydroxyl and amino groups contained in PVA and cPAM. In addition, due to the addition of cPAM, the zeta potential of the PVC biofilm carrier gradually increased. The surface chargeability of PVC gradually changes from negative to positive. When 5 wt % PVA and 15 wt % cPAM were added, the surface contact angle and zeta potential of PVC are 81.6° and 14.6 mV. In addition, the holes clearly appeared on the surface of PVC after 1 wt % mAC was added. This result shows that the hydrophilicity, electrophilicity, and specific surface area of the PVC biofilm carrier have been significantly improved. Therefore, it is beneficial to the adsorption of bacteria and microorganisms.

## Figures and Tables

**Figure 1 polymers-12-01240-f001:**
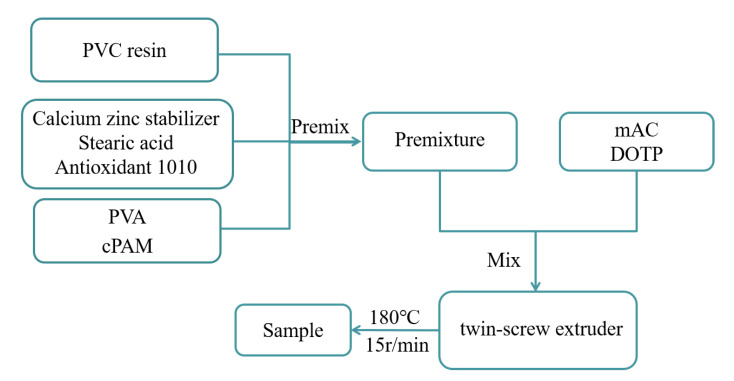
Preparation of PVC biofilm carrier.

**Figure 2 polymers-12-01240-f002:**
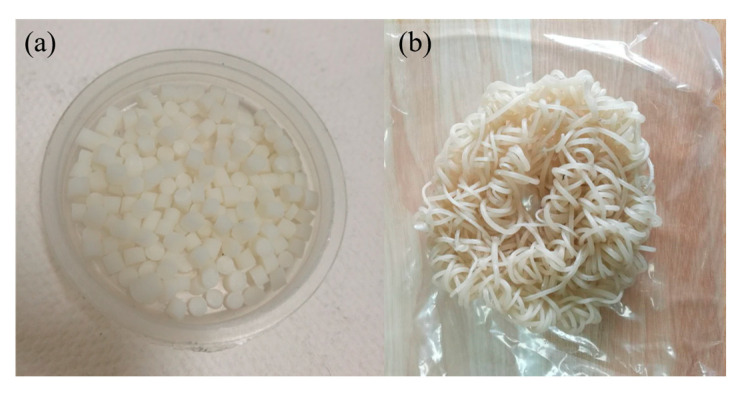
The photo of sample (**a**) sample particles, (**b**) sample.

**Figure 3 polymers-12-01240-f003:**
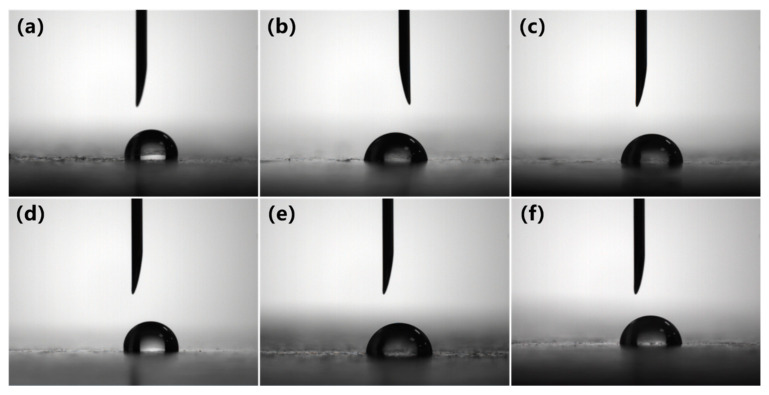
Surface contact angle images of the PVC biofilm carrier: (**a**) PVC, (**b**) PVC/PVA-5, (**c**) PVC/PVA-10, (**d**) PVC/mAC-PVA, (**e**) PVC/mAC-PVA-5, (**f**) PVC/mAC-PVA-10.

**Figure 4 polymers-12-01240-f004:**
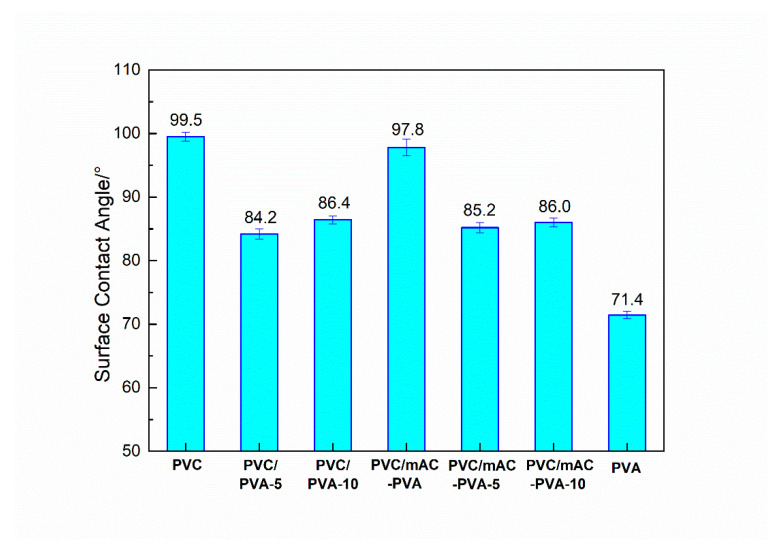
Surface contact angle of the PVC biofilm carrier.

**Figure 5 polymers-12-01240-f005:**
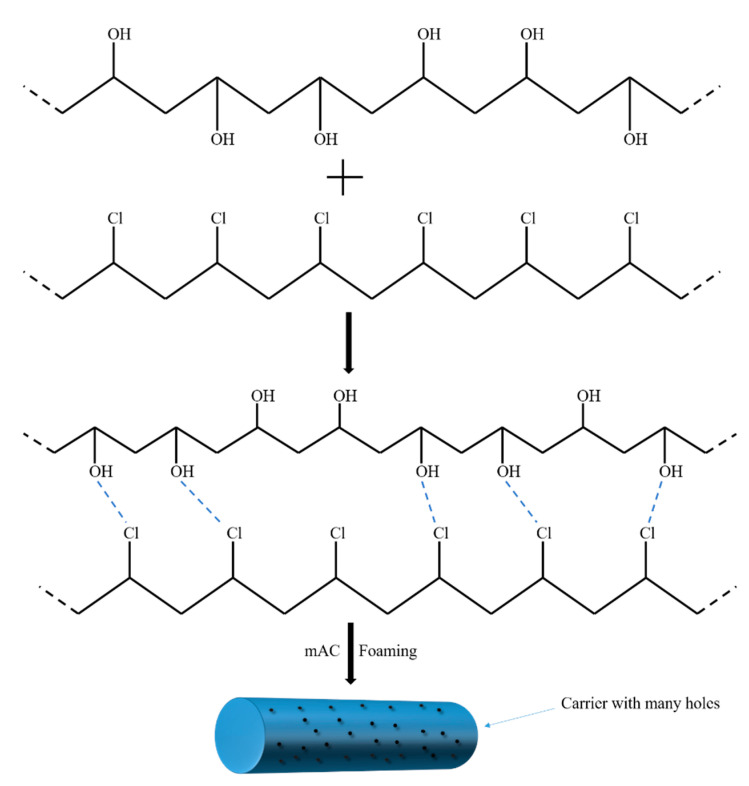
The reaction mechanism of PVC’s modification.

**Figure 6 polymers-12-01240-f006:**
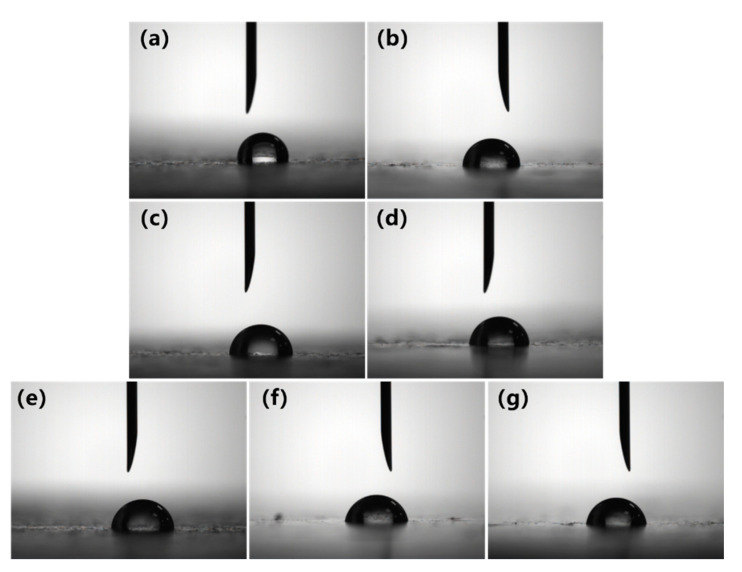
Surface contact angle images of the PVC modified by cPAM: (**a**) PVC, (**b**) PVC/cPAM, (**c**) PVC/cPAM-3, (**d**) PVC/cPAM-6, (**e**) PVC/cPAM-9, (**f**) PVC/cPAM-12, and (**g**) PVC/cPAM-15.

**Figure 7 polymers-12-01240-f007:**
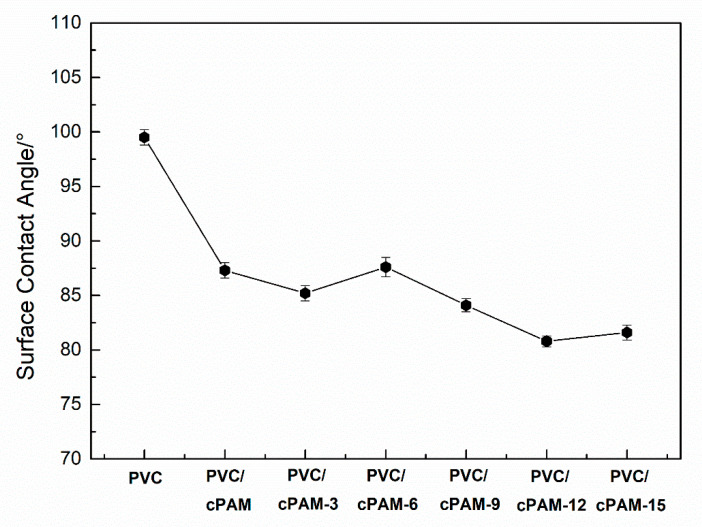
Surface contact angle of the PVC modified by cPAM.

**Figure 8 polymers-12-01240-f008:**
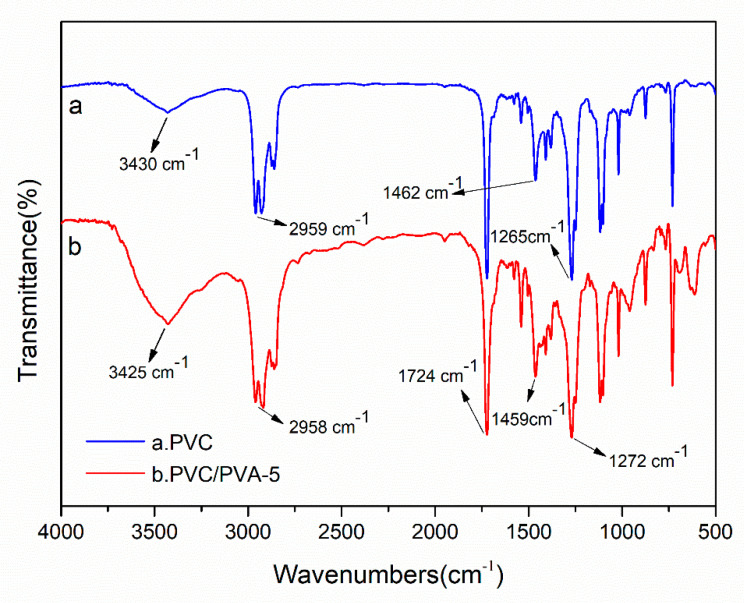
Fourier transform infrared (FTIR) spectra of (**a**) PVC and (**b**) PVC/PVA-5.

**Figure 9 polymers-12-01240-f009:**
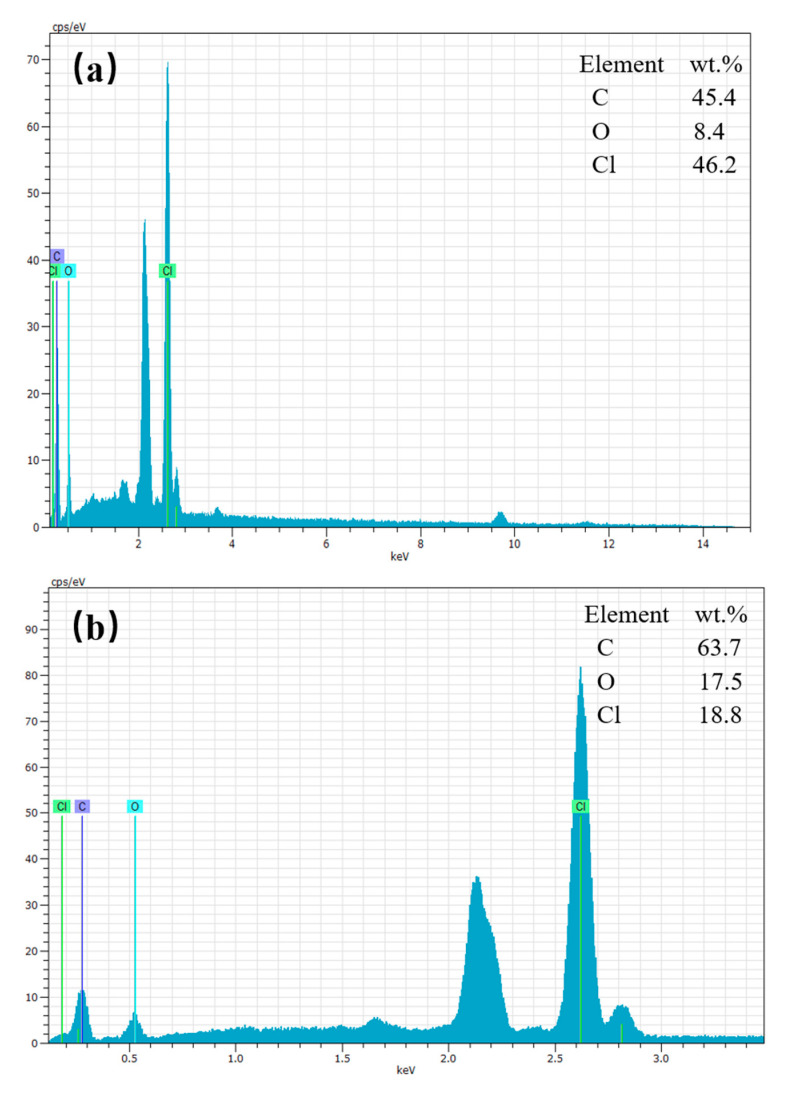
EDS spectra of (**a**) PVC and (**b**) PVC/PVA-5.

**Figure 10 polymers-12-01240-f010:**
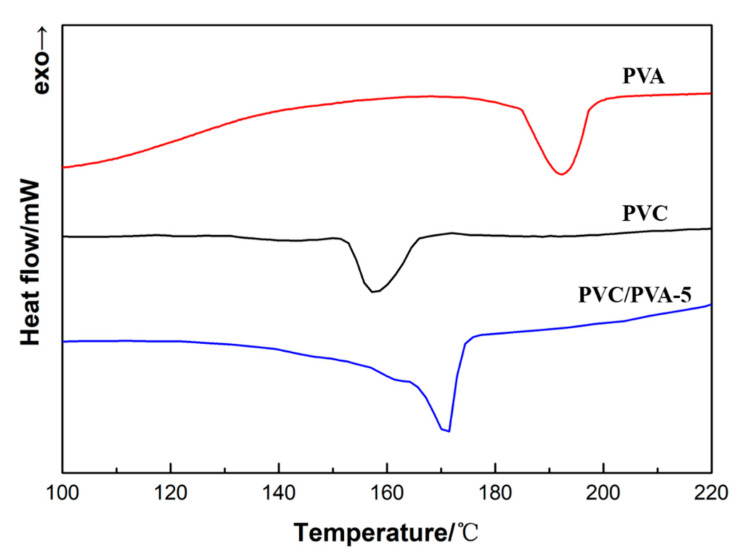
The melting characteristics of PVC, PVA, and PVC/PVA-5.

**Figure 11 polymers-12-01240-f011:**
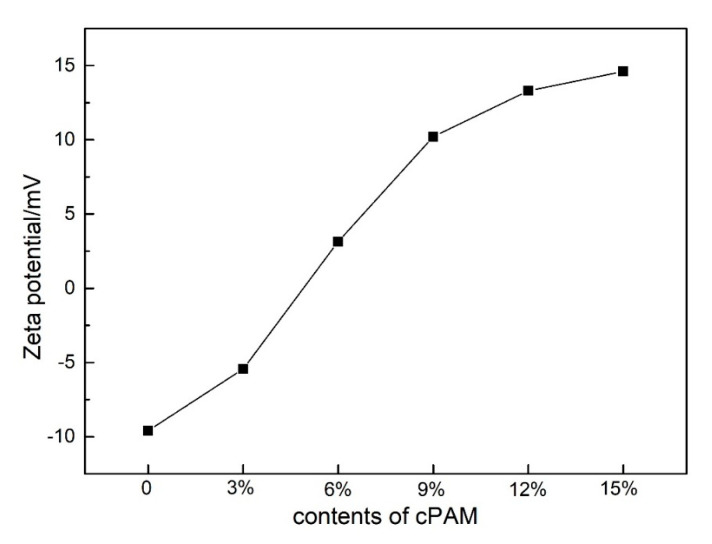
Zeta potential of the PVC biofilm carrier.

**Figure 12 polymers-12-01240-f012:**
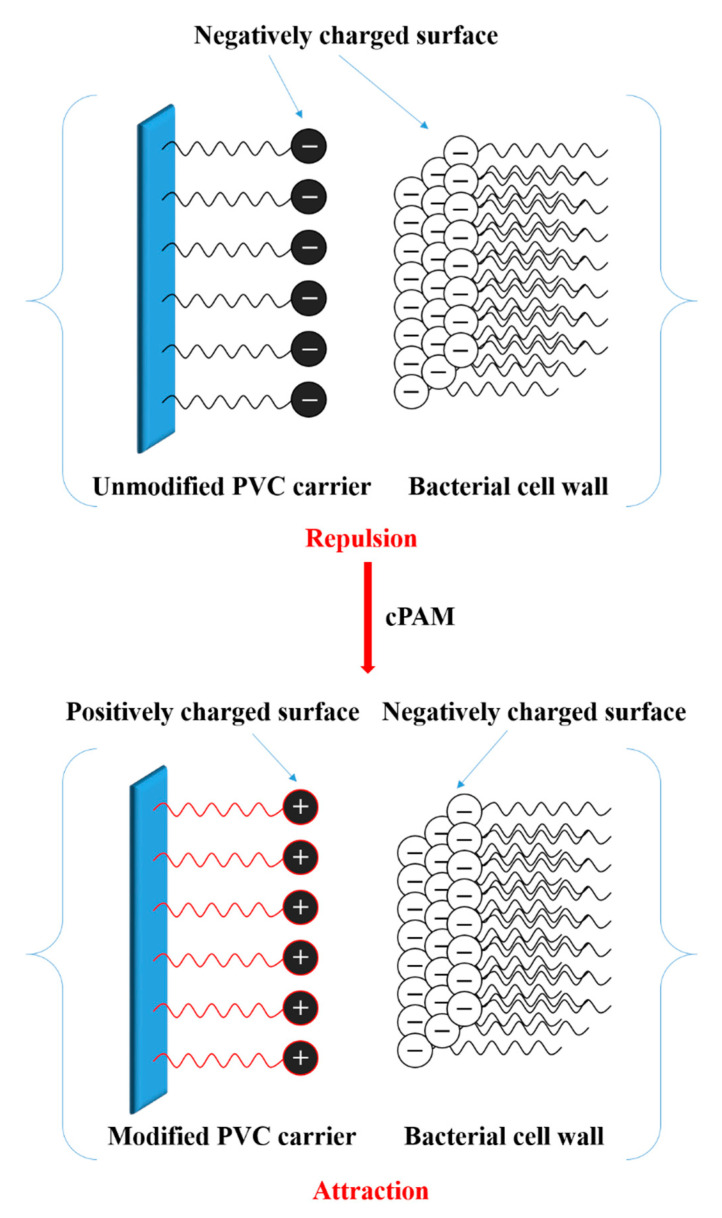
The electrostatic interactions between unmodified PVC/modified PVC and bacteria.

**Figure 13 polymers-12-01240-f013:**
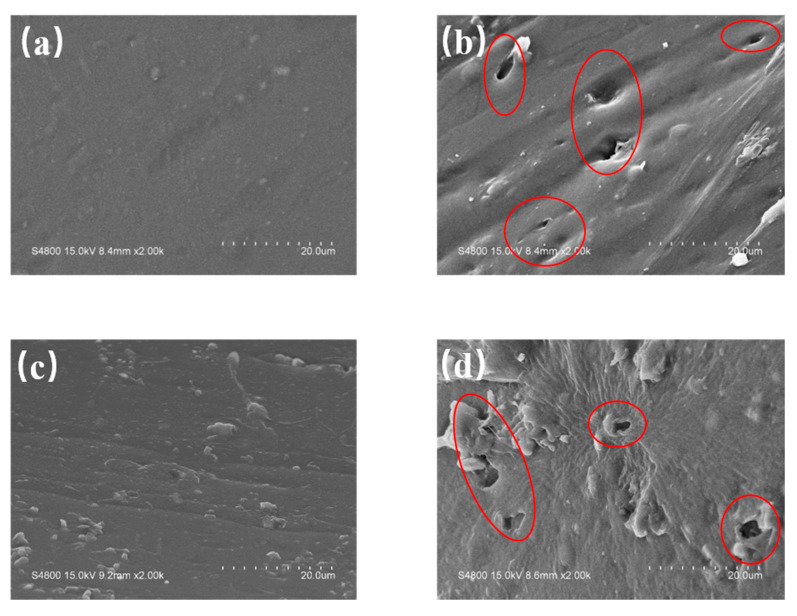
SEM images of surface morphology of the PVC biofilm carrier: (**a**) PVC, (**b**) PVC/mAC-PVA, (**c**) PVC/PVA-5, (**d**) PVC/mAC-PVA-5. The red circle is the hole.

**Figure 14 polymers-12-01240-f014:**
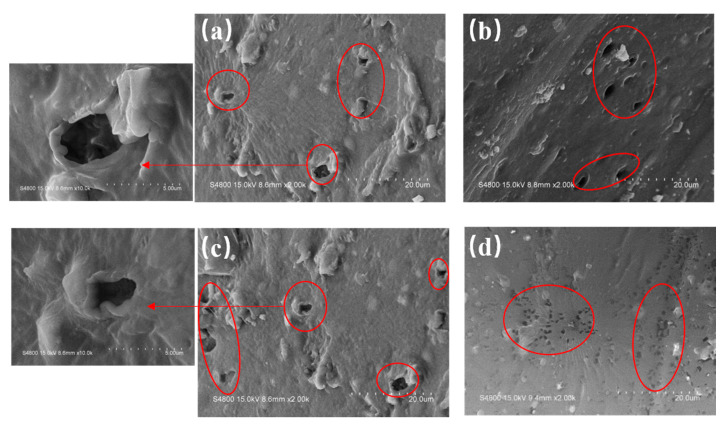
SEM images of surface morphology of the PVC biofilm carrier: (**a**) PVC/cPAM, (**b**) PVC/cPAM-3, (**c**) PVC/cPAM-12, and (**d**) PVC/cPAM-15. The red circle is the hole.

**Figure 15 polymers-12-01240-f015:**
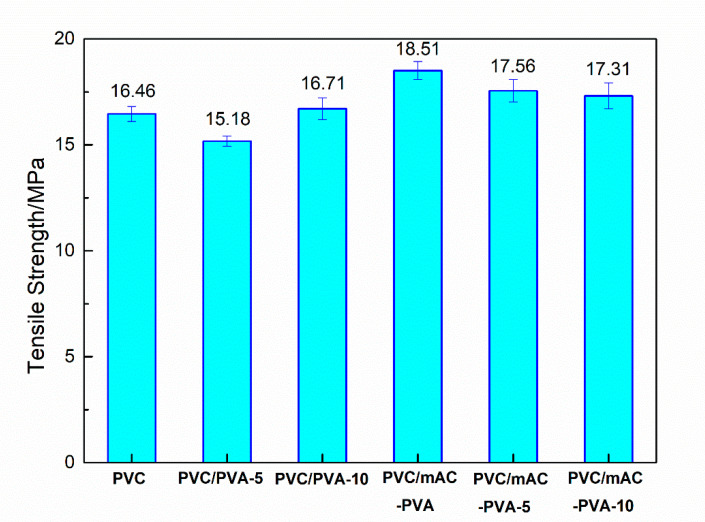
Tensile strength of the PVC biofilm carrier.

**Figure 16 polymers-12-01240-f016:**
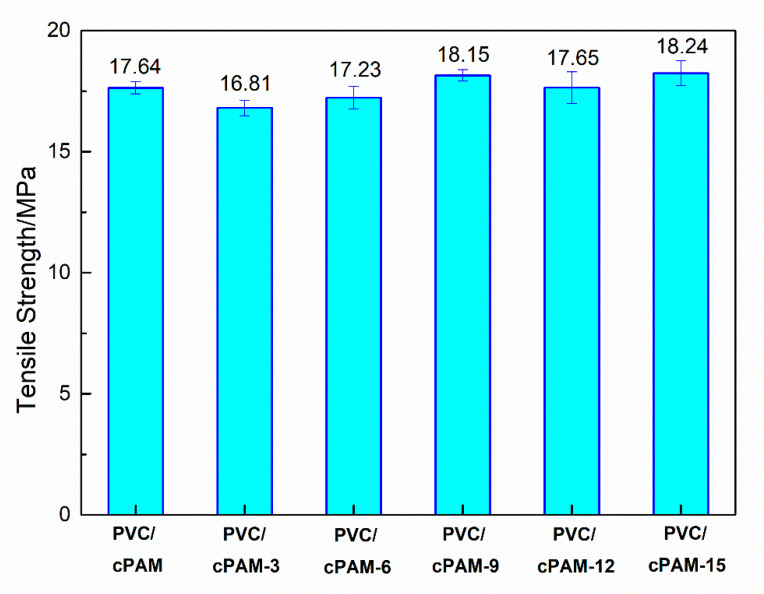
Tensile strength of the PVC modified by cPAM.

**Table 1 polymers-12-01240-t001:** The information of the materials. cPAM: cationic polyacrylamide, PVA: polyvinyl alcohol, PVC: Polyvinyl chloride.

Material	Sales/Manufacturer	Material Properties
PVC (SG-5)	Yuyao Maiduo Plastic Chemical Co., Ltd.	GB/T 5761-2006 Injection Grade
PVA	Shanghai Lingfeng Chemical Reagent Co., Ltd.	A.R, Average degree of polymerization:1750 ± 50
Stearic acid	Shanghai Lingfeng Chemical Reagent Co., Ltd.	A.R, MW:284.48
cPAM	Zhengzhou Jintai Environmental Protection Technology Co., Ltd.	A.R, MW:1.2 × 10^7^
Azodicarbonamide	Shanghai Tengzhun Biological Technology Co., Ltd.	A.R, MW:116.08
Calcium zinc stabilizer	Guangdong Winner New Material Technology Co., Ltd.	WWP-F02 A.R
Dioctyl terephthalate	Jining Baichuan Chemical Co., Ltd.	A.R
Zinc oxide	Sinopharm Chemical Reagent Co., Ltd.	A.R, MW:81.39
Antioxidant 1010	Shanghai Xian Ding Biological Technology Co., Ltd.	C.P, MW:1177.63

**Table 2 polymers-12-01240-t002:** The formulation and abbreviation of the samples.

Samples	PVC/g	PVA/g	mAC/g	cPAM/g
PVC	100	0	0	0
PVC/PVA-5	100	5	0	0
PVC/PVA-10	100	10	0	0
PVC/mAC-PVA	100	0	1	0
PVC/mAC-PVA-5	100	5	1	0
PVC/mAC-PVA-10	100	10	1	0
PVC/cPAM	100	5	1	0
PVC/cPAM-3	100	5	1	3
PVC/cPAM-6	100	5	1	6
PVC/cPAM-9	100	5	1	9
PVC/cPAM-12	100	5	1	12
PVC/cPAM-15	100	5	1	15

**Table 3 polymers-12-01240-t003:** The EDS results of PVC and PVC/PVA-5.

Sample	PVC		PVC/PVA-5	
	Weight (wt %)	Atom (at. %)	Weight (wt %)	Atom (at. %)
C	45.4 ± 2.4	67.4 ± 2.1	63.7 ± 4.2	76.6 ± 4.0
O	8.4 ± 0.5	9.3 ± 0.4	17.5 ± 1.4	15.8 ± 1.2
Cl	46.2 ± 0.6	23.3 ± 0.5	18.8 ± 0.3	7.6 ± 0.2

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
