# Peer review of "Enhanced Hydrophilic and Electrophilic Properties of Polyvinyl Chloride (PVC) Biofilm Carrier"

_polymers, 2020, doi:10.3390/polym12061240_

Round 1
Reviewer 1 Report
The manuscript was re-submitted after a good revision. The revised paper has improved and I think that publication of the manuscript at this point is possible.
Author Response
Thanks for your suggestion.
Reviewer 2 Report
Most of the comments have been included in the latest version of the article. The manuscript can be considered as publication material.
Author Response
Thanks for your suggestion.
Reviewer 3 Report
This manuscript has presented intersting multi-step methodology for the preparation of the bacteria-adhesion membranes based on PVC. Authors have shown the process allowed for changing Zeta potential of the polymer leading to positive poteintial from negative to positive that enhanced the ability for bacteria adhsion. Although this work is novel and quite well organized, some improvements are necessary before to proceed further.
1) FT-IR. Authors should carrefully analyze FT-IR data. All imporant peaks must defined and assingned on the Figures.
2) SEM data. Authors claimed higher porosity due to some holes that were formed during processing. However, I can't see clearly these claimed holes. The holes that were shown on single images seemed as unhomogeneity of the polymer. Thus, authors should present better pictures or show more pictures in the Supporting material at different magnifications. The best would be to combine it with other techniques e.g. AFM.
Round 2
Reviewer 3 Report
Authors have significantly improved the quality of this paper. Authors improved FT-IR analysis and changed SEM micrographs.
I do recommend to accept this mananuscript for publication.
This manuscript is a resubmission of an earlier submission. The following is a list of the peer review reports and author responses from that submission.
Round 1
Reviewer 1 Report
nice paper!
I think the publication is ready for publication.
Author Response
Thank for your suggestion.
Reviewer 2 Report
The manuscript prepared by the authors summarized the results of their research regarding the use of modified PVC material as the biofilm carier. In my opinion the presented research topic is not novel, also the research results should be supplemented with several important aspects.
There are no sample photos in the paper, which would indicate whether the obtained mixtures are transparent or not.
The SEM pictures are presenting only some general view on the foil surface, without any structure analysis.
What is the purpose of presenting EDS results, they do not present any new conclusions.
Without bacteriological studies, all conclusions regarding favorable structure properties after modification are unreliable
Other comments:
The description of the film preparation procedure is very short.
The results of contact angle for PVA and PAM modified PVC are presented inconsistently.
In principle, the same note applies to the whole section of characterization methodology, there is lack of parameters and test conditions.
Reviewer 3 Report
The paper presents experimental results of some technological interest, however, in my opinion, the paper in the actual form is inadequate for deserving publication in Polymers. First of all, the manuscript must be strongly revised, trying to clarify many sentences.
In addition, I advise the authors to review the manuscript based on the following points:
- This subject is not new. It is important to underline the original contribution of the authors. Please compare these results with previous studies and report their new insights in this field.
- It is mandatory to improve the quality and the consistency of the paper by adding more information on the material used. In particular, a better morphological characterization of the film is necessary. It is necessary to confirm the dispersion and compatibility of the blends.
- Surface contact angle results are quite strange. In any case, explain the reason for such behavior and, if possible, add references with similar results.
- The list of references seems skimpy; it does not include recent papers in the field.
Round 2
Reviewer 2 Report
The authors have completed the manuscript with most of the information required in the review. In its current form, the article may have been taken into account as a publication in the Polymers journal.
Author Response
Thanks for your suggestion.
Reviewer 3 Report
The manuscript was re-submitted after a good revision. However, as in my first comment, I think the document still has some weaknesses that need to be reviewed before publication. The explanation of the contact angle behavior is rather strange. I can't understand why, with increasing PVA content, the value of the contact angle increases.
- What is the contact angle of the PVA? (please add it in figure 4)
- What is the role of the surface topology of the samples (micro holes, microstructures, partial separation of the components)?
- Are PVA and PVC completely miscible or not? What is the miscibility percentage of the two polymers?
- Please add further tests to evaluate the miscibility of the two polymers in the present samples (DSC, rheology, DMA, etc.)
- What is the role of mAC and cPAM in these mixtures?
- What is the contact angle of cPAM?
- Are PVC and cPAM miscible?
- Are the samples really homogeneous? How was this aspect assessed?
Round 3
Reviewer 3 Report
Dear author, I understand your current difficulties very well. Unfortunately, they are a common problem for everyone.
I thank you for having answered my questions in a timely manner, but I ask you to insert the text of your answers also in the manuscript.
You have to understand that the questions a reviewer asks you are the same as any reader of your article would. If the doubt has come to the reviewer, it will also come to the reader. It is the duty of the author to try to help the reader understand. This increases the chance of science progressing and also helps your article increase visibility.
I ask you the question again. Why does the PVC / PVA-5 sample have a smaller contact angle than the PVC / PVA-10 sample? It seems that with the increase of the PVA content the contact angle increases.
The work reads: "The reason for this phenomenon is that the amount of PVA added is too large, causing the hydroxyl group between PVA to act to generate hydrogen bonds, which in turn reduces the binding of PVA to PVC and the number of PVA independent hydroxyl groups"
I don't agree. So, try to find a reference in the literature that supports this observation. Above all, use the conditional verbs when making observations of which you have no experimental evidence: "This could be due to ..."